# Effects of Probiotic Supplementation on Exercise with Predominance of Aerobic Metabolism in Trained Population: A Systematic Review, Meta-Analysis and Meta-Regression

**DOI:** 10.3390/nu14030622

**Published:** 2022-01-30

**Authors:** Asier Santibañez-Gutierrez, Julen Fernández-Landa, Julio Calleja-González, Anne Delextrat, Juan Mielgo-Ayuso

**Affiliations:** 1Physical Education and Sports Department, Faculty of Education and Sport, University of the Basque Country (UPV/EHU), 01007 Vitoria, Spain; asantibanez004@ikasle.ehu.eus (A.S.-G.); julenfdl@hotmail.com (J.F.-L.); julio.calleja.gonzalez@gmail.com (J.C.-G.); 2Department of Sport and Health Sciences, Oxford Brookes University, Oxford OX3 0BP, UK; adelextrat@brookes.ac.uk; 3Department of Health Sciences, Faculty of Health Sciences, University of Burgos, 09001 Burgos, Spain

**Keywords:** probiotics, supplementation, physical performance, aerobic, recovery

## Abstract

The scientific literature about probiotic intake and its effect on sports performance is growing. Therefore, the main aim of this systematic review, meta-analysis and meta-regression was to review all information about the effects of probiotic supplementation on performance tests with predominance of aerobic metabolism in trained populations (athletes and/or Division I players and/or trained population: ≥8 h/week and/or ≥5 workouts/week). A structured search was performed in accordance with the Preferred Reporting Items for Systematic Reviews and Meta-Analyses (PRISMA^®^) statement and PICOS guidelines in PubMed/MEDLINE, Web of Science (WOS), and Scopus international databases from inception to 1 November 2021. Studies involving probiotic supplementation in trained population and execution of performance test with aerobic metabolism predominance (test lasted more than 5 min) were considered for inclusion. Fifteen articles were included in the final systematic review (in total, 388 participants were included). After 3 studies were removed due to a lack of data for the meta-analysis and meta-regression, 12 studies with 232 participants were involved. With the objective of assessing the risk of bias of included studies, Cochrane Collaboration Guidelines and the Physiotherapy Evidence Database (PEDro) scale were performed. For all included studies the following data was extracted: authors, year of publication, study design, the size of the sample, probiotic administration (dose and time), and characteristics of participants. The random effects model and pooled standardized mean differences (SMDs) were used according to Hedges’ g for the meta-analysis. In order to determine if dose and duration covariates could predict probiotic effects, a meta-regression was also conducted. Results showed a small positive and significant effect on the performance test with aerobic metabolic predominance (SMD = 0.29; CI = 0.08–0.50; *p* < 0.05). Moreover, the subgroup analysis displayed significant greater benefits when the dose was ≥30 × 10^9^ colony forming units (CFU) (SMD, 0.47; CI, 0.05 to 0.89; *p* < 0.05), when supplementation duration was ≤4 weeks (SMD, 0.44; CI, 0.05 to 0.84; *p* < 0.05), when single strain probiotics were used (SMD, 0.33; CI, 0.06 to 0.60; *p* < 0.05), when participants were males (SMD, 0.30; CI, 0.04 to 0.56; *p* < 0.05), and when the test was performed to exhaustion (SMD, 0.45; CI, 0.05 to 0.48; *p* < 0.05). However, with references to the findings of the meta-regression, selected covariates did not predict probiotic effects in highly trained population. In summary, the current systematic review and meta-analysis supported the potential effects of probiotics supplementation to improve performance in a test in which aerobic metabolism is predominant in trained population. However, more research is needed to fully understand the mechanisms of action of this supplement.

## 1. Introduction

Probiotics are “Live microorganisms, which administered in adequate amounts, confer benefits on the health of the host” [1]. Although there are many bacteria used as probiotics, the most commonly used strains belong to *Bifidobacterium, Lactobacillus*, *Enterococcus*, or *Propionibacterium* genera or certain yeasts, for example, *Saccharomyces boulardii* [2]. Even though they may exist within the same species, two different probiotic strains possess unique transcriptomes with different mechanisms of action and potential benefits [3]. Among other physiological functions, probiotics can influence the immune functions, help to maintain a healthy digestive system, avoid respiratory illness or persistent common cold and reduce inflammation [4]. In particular, the mechanisms by which probiotics improve health are: increased intestinal barrier function of the epithelial cells [5]; modification of macrophage/lymphocyte cytokines secretion [6]; antibacterial effects on colonization [7]; regulation of the production of antimicrobial peptides and antioxidant compounds/enzymes [8]; induction of T cells regulation [9]; increased communication between the immune system and the microbiota [10] and involvement of short-chain fatty acids (SCFAs) in the homeostasis of regulatory T lymphocyte [11].

Moreover, probiotics can also reduce upper-respiratory-tract infections (URTIs), gastrointestinal disorders (GI), and oxidative stress, which compromise an athlete’s health status [12]. In this sense, intense and/or continuous practice could stress athletes, inducing them to several health complications such as immunity depression, inflammatory dysregulation, increased URTIs, increased oxidative and mental stress, GI symptoms, and endotoxemia [13] that if not resolved could compromise their performance [14,15,16]. In that way, a recent systematic review and meta-analyses carried out on professional athletes shows the effectiveness of probiotic supplementation in decreasing the total severity score of URTIs [17]. In addition, the authors found a decrease in two inflammatory cytokines, interleukin-6 (IL-6) and tumor necrosis factor alfa (TNF-α) levels. Moreover, some probiotic strains could reduce exercise overproduction of reactive oxygen species (ROS) by their antioxidant effect [18,19]. Concretely, Zamani et al. recently conducted a systematic review and meta-analysis and found that probiotic supplementation significantly augmented total antioxidant capacity (TAC) and reduced malondialdehyde (MDA), an oxidative marker [20].

On the other hand, apart from influencing health status, probiotics could also affect physical performance, in particular in exercise in which the aerobic metabolism is predominant (≥5 min) [21]. Based on a previous study conducted by our research group as this study, tests used to measure sports performance were classified as aerobic when the test lasted more than 5 min (≥5 min) [22]. In this sense, prolonged exercise, leads to a splanchnic hypoperfusion that stresses the GI tract, and this increases gut permeability and therefore allows endotoxemia to occur [23]. In addition, the GI disorders are common among endurance athletes—30–50% could suffer GI complaints [23]. Consequently, probiotic supplementation could help to improve the intestinal barrier, and avoid endotoxemia and the following process of inflammation [5]. Moreover, several probiotic strains have shown their capability to digest, absorb and metabolize important nutrients for sports performance and recovery [24]. In this sense, probiotics supplementation has enhanced the amino acid absorption from plant protein [24] and has increased exogenous glucose absorption and oxidation during exercise [25]. Another mechanism of probiotics supplementation to improve aerobic capacity could be through the production of SCFAs, which are an extra energy source during exercise [26]. In addition, some SCFAs could increase peroxisome proliferator activated receptor (PGC-1α) and hence, the proportion of type I fibers [27].

As mentioned before, probiotics supplementation could influence aerobic metabolism positively due to the different physiological pathways. However, research on the impact of probiotics on exercise with aerobic metabolism predominance has yielded conflicting results so far due in part to a wide variety of methodologies used.

In this sense, a wide range of protocols are used [4]. Therefore, it is difficult to standardize a supplementation protocol in order to increase performance as some studies, for example, were conducted after athletes consumed probiotic supplementation without taking any food, others when food was eaten and others after exercise [4]. In addition, while some studies used a single bacteria probiotics strain [28,29,30,31,32,33,34,35,36], others used different probiotic strains [5,25,37,38,39,40]. Moreover, most of the research measuring probiotic effects have been carried out in males [5,25,28,30,33,38,40]. Therefore, the main aims of this systematic review and meta-analysis were on one hand to evaluate the effect of probiotic supplementation on exercise in which the aerobic metabolism is predominant (≥5 min) in a trained population and, on the other hand, to determine the doses and time of treatment of probiotic supplementation and to investigate the differences between sexes and probiotic effectiveness in a highly trained population. In addition, it could be useful to study the differences between single strain and multi strain probiotics, and variances concerning the test carried out (measuring VO_max_ and test performed to fatigue).

## 2. Material and Methods

### 2.1. Literature Search Strategies

A systematic search of the scientific literature was carried out in accordance with PRISMA^®^ (Preferred Reporting Items for Systematic Reviews and Meta-Analyses) statement guidelines [41] to investigate the effects of probiotic supplementation on sports with aerobic metabolism predominance. A systematic search of the current scientific literature was carried out for studies that investigated the supplementation of probiotics on aerobic metabolism capacity in a trained population in a competitive environment. The PICOS model for the definition of the inclusion criteria was followed: P (Population): “athletes and/or Division I and/or trained population (≥8 h/week and/or ≥5 workouts/week)”; I (Intervention): “effects of probiotic supplementation on test with predominance on aerobic metabolism”; C (Comparators): “similar experimental conditions in the placebo or control group compared with the probiotic group”; O (Outcome): “performance test with aerobic metabolism dominance”; and S (Study design): “double-blind controlled clinical trial” [42].

Records were identified by searching in PubMed/Medline, Web of Science (WOS), and Scopus from inception to 1 November 2021. For that purpose, the following Boolean search equation was used for the PubMed/MEDLINE database: (“probiotics” [MeSH Terms] OR “probiotics” [All Fields]) AND ((“exercise” [MeSH Terms] OR “exercise” [All Fields]) OR (“sports” [MeSH Terms] OR “sports”[All Fields] OR “sport”[All Fields]) OR performance[All Fields] OR aerobic[All Fields]) AND ((“athletes” [MeSH Terms] OR “athletes” [All Fields] OR “athlete” [All Fields]) OR trained[All Fields] OR elite[All Fields]). For the WOS and Scopus databases, the following Boolean search was used: (Probiotics and (exercise or sport or performance or aerobic) and (athlete or trained or elite)) (further information could be found in Appendix A).

Apart from this search, other articles were included using the snowball strategy (*n* = 4) [43]. All titles and abstracts were search cross-referenced in order to find duplicates and other possible missing studies and were then screened for a full-text review. Two authors (A.S.-G. and J.F.-L.) independently performed the search for published studies. Disagreements were solved through discussion with a third author (J.C.-G.).

### 2.2. Inclusion Criteria and Exclusion Criteria

For the articles obtained in this systematic review and meta-analysis, the next inclusion criteria were applied to choose studies: (i) a well-designed experiment; (ii) used test in which aerobic metabolism is primary; (iii) participants had to be athletes and/or Division I and/or trained population (≥8 h/week and/or ≥5 workouts/week clinical trial); (iv) a clinical trial; (v) with clear information concerning supplementation administration; (vi) peer-reviewed and original articles written in the English language; and (vii) clear information about funding sources. In addition to those criteria, studies were excluded if there was unclear information concerning probiotic supplementation, and if participants had previous health problems or injuries leading to drug intake. Table 1 displayed more details concerning inclusion and exclusion criteria for included studies.

### 2.3. Text Screening

Once the inclusion/exclusion criteria had been applied to each study, the data on the study source (including the authors and publication year), the characteristics of the participants (level, and sex), the study design, the way of administration of the supplement (dose and time), and sample size were extracted.

Two investigators (A.S.-G. and J.F.-L.) independently screened titles and abstracts of the initial search results based upon a priori inclusion and exclusion criteria using a spreadsheet (Microsoft Inc.^®^, Seattle, WA, USA). Subsequently, full texts were independently screened by the same two investigators (A.S.-G. and J.F.-L.) to determine which studies warranted inclusion in this analysis. Any disagreement between these two investigators (A.S.-G. and J.F.-L.) was resolved through discussion or using third-party adjudication (J.C.-G.).

### 2.4. Data Extraction, Study Coding, and Quality Assessment

All studies meeting the inclusion criteria were reviewed and data were recorded in a spreadsheet (Microsoft Excel, Microsoft Corporation^®^, Washington, DC, USA). This information included study authors; study design; year of publication; participants’ ages; participants’ sex; participants’ training status, dose and type of supplementation; supplementation timing and exercise outcomes of the intervention. In sports performance variables, performance was considered predominantly aerobic if the test lasted longer than 5 min [21].

Risk of bias figures were performed with Review Manager (Revman) v5.3^®^ (Copenhagen: The Nordic Cochrane Centre, The Cochrane Collaboration, 2014). Comprehensive Meta-Analysis program^®^ (v2.0; Biostat, Englewood, NJ, USA) was used to performed Egger’s statistic test with the aim of detecting publication bias, where *p* ≤ 0.05 was considered bias. Egger´s analyses suggested that no publication bias was found on aerobic performance (z = 1.33; *p* = 0.11). Funnel plots are presented in Figure 1.

The quality assessment of the included studies was evaluated by 2 investigators (A.S.-G. and J.F.-L.). Quality assessment was carried out in accordance with the Cochrane Collaboration Guidelines [44], which divided quality and risk of bias into 6 domains: selection bias; performance bias; detection bias; attrition bias; reporting bias, and other types of bias. A domain is considered as “low risk” of bias if possible bias is unlikely to seriously alter criteria results, or as “high risk” when probable bias seriously weakens confidence in the results; or it could be considered as “unclear” when there was plausible bias that raises some doubt about the results. In addition, the Physiotherapy Evidence Database (PEDro) scale [45] was used with the aim to assess the methodological quality of included studies, in which quality assessment is divided into 11 items: eligibility criteria; random group allocation; concealed allocation; similar groups at baseline; blinding of participants; blinding of coaches; blinding of assessors; 85 % of participants received at least 1 key measurement; intention to treat; between-group statistical comparison reported for at least 1 key outcome; and effects sizes and measures of variability [45]. The maximum score for each item is 10. Details of each article and domains are presented in Figure 2 and Figure 3 and Table 2. The study protocol was registered in the Prospective Register of Systematic Review (PROSPERO) with the following registration number: CRD42021248173.

### 2.5. Statistical Analysis

Review Manager (Revman) v5.3 (Copenhagen: The Nordic Cochrane Centre, The Cochrane Collaboration, 2014) was also used for descriptive analyses and meta-analytic statistics. For the statistical analysis sample sizes, group means and standard deviations (SD) were extracted for the different outcomes in the group supplemented with probiotics and in the placebo pre- and post-treatment. When there were no numerical values and data were presented as figures, values were estimated based on pixel count using calibrated images in Image J software (National Institutes of Health, Bethesda, MD, USA).

In order to contrast the ingestion of probiotics vs. placebo, the number of participants, the standardized mean difference (SMD), and standard error of the SMD were calculated for each measured outcome meeting inclusion criteria. Hedges’ g was used to calculate SMD of probiotic and placebo groups [46]. Weighting SMD by the inverse of variance and overall effect and its 95% confidence interval (CI) was performed. Furthermore, both group´s SMDs were utilized to get the net treatment effect, and pooled SD of changes scores were used to calculate variance. The DerSimonian and Laird method [47] was used for the random effects model. The magnitude of the SMD effect was interpreted as: trivial if it was <0.2; small if between 0.2–0.3; moderate if 0.4–0.8; and large if >0.8 following the Cohen criteria [48].

Statistic *I*^2^ was used to estimate statistical heterogeneity across the included trials [44]. For heterogeneity, *I*^2^ values range from 0 to 100%. Thus, between 25% and 50% indicates a small risk of heterogeneity, between 50% and 75% represents a medium risk of heterogeneity, and higher than 75% indicates a large risk of heterogeneity [49].

A multivariate random-effects meta-regression was performed with the aim of verifying whether any of the covariables (dose and duration) predicted probiotic effects on performance test in which aerobic metabolism is predominant in a highly trained population. Meta-regression analysis was conducted with Open Meta-Analyst software^®^.

## 3. Results

### 3.1. Main Research

The initial literature search through the previous selected databases yielded a total of 692 records. Among them, 502 were single records and 186 were duplicates. In addition, 4 additional records were included by reference list searches with a total number of 506 articles identified. Titles and abstract screening removed 474 studies: 15 were non-human studies, 69 were regarding disease treatment, 60 were reviews, and 330 studies were eliminated for other reasons (i.e., not dealing with probiotics and performance). Only 32 eligible studies were assessed for full-text screening. Among them, 17 studies were excluded for different reasons (in 5 studies the population was not considered trained people, in 8 studies sports performance was not measured, and 4 did not measure aerobic performance, more information could be found in Appendix A). Finally, 15 studies [5,25,28,29,30,31,32,33,34,35,36,37,38,39,40] were included in this systematic review. Then 3 studies were removed, 2 because data regarding the performed test were not shown [31,34], and the remaining study due to insufficient reported data (only baseline data was shown [39]), leaving 12 articles definitively included in this meta-analysis. For more visual understanding, the PRISMA flow diagram is presented in Figure 4.

### 3.2. Probiotic Supplementation

In total, 388 participants were included in the systematic review and 232 in the final meta-analysis. Regarding the used probiotic strains, 9 studies used a single strain [28,29,30,31,32,33,34,35,36], whereas in 6 studies participants consumed a multi strain probiotic [5,25,37,38,39,40]. Concerning probiotic supplementation, 11 studies used capsules [25,28,29,30,31,34,35,36,37,38,40], 2 utilized powder sachets [5,39], 1 study used tablets [38], and the other one used a probiotic drink. As to supplement duration, there was a large variation ranging from 3 weeks [29] to 14 weeks [5]. Regarding the supplementation dose used, this was diverse across studies, going from 1.0 × 10^9^ [34,35,39] to 4.5 × 10^10^ colony-forming units (CFU) [38]. Supplementation timing was just as varied: before [39] or after the first meal [25,31]; during any meal [5,28] or during the 3 main meals [36]; throughout the day with or without a meal [34]; or after the exercise session and before sleeping [29,30]. Other studies lacked information about the timing of supplementation [32,33,35,37,38,40]. Concerning the strains belonging to *Lactobacillus* genera, *L. fermentum* VRI-003PCC^®^, *L. fermentum*; *L. plantarum* PS1228, *L. plantarum; L. helvelticus* Lafti^®^10; *L. acidophilus* CUL-60, *L. acidophilus* CUL-61, *L. acidophilus* W22, *L. acidophilus*; *L. casei*; heat-killed of *L. gasseri* OLL2809; *L. brevis* W63 and *L. rhamnosus* were consumed. Upon *Bifidobacterium* genera, *B. longum* 35624, *B. longum* R0175, *B. bifidum*, *B. bifidum* W23, *B. bifidum* (CUL-20), *B. lactis*, *B. lactis* W51, *B. lactis* CUL-34, *B. lactis* Lafti B94 and *B. breve* were used. Other strains were also utilized, such as *Enterococcus faecium* W54 and *E. faecium* R0026, *Bacillus subtilis* R0179 and *Streptococcus thermophilus*. Regarding sex, most studies included in the systematic review were performed in males (*n* = 8) [5,25,28,30,32,33,38,40]. A single study was conducted in women [35], while five studies mixed both sexes [31,34,36,37,39]. The remaining study did not define the sex of the athletes [29].

### 3.3. Effect of Probiotics on Exercise in Which Aerobic Metabolism Is Predominant (≥5 min)

Table 3 and Table 4 show the tests carried out to assess performance. Significant improvements were observed in 5 studies [29,30,32,36,38]. In 3 trials [29,30,38], the improvement was shown in time to exhaustion. The remaining studies observed an increment in distance performed in a 12 min running/walking Cooper test [36] and 20 m multi-stage run test [32].

Additional information concerning design of the studies and nutritional control aspects in participants are shown in Table 5.

### 3.4. Effect of Probiotics on Exercise in Which Aerobic Metabolism Is Predominant (≥5 min). Meta-Analysis

Figure 5 displays a small and significant effect on performance tests in which aerobic metabolism is primary (SMD = 0.29; 95% IC = 0.08–0.50; *p* < 0.05) after following probiotic supplementation. In addition, this meta-analysis reported small heterogeneity among studies reviewed (*I*^2^ = 14%; *p* = 0.29). In particular, Shing et al. [38] and Huang et al. [30] observed a large positive effect in time to fatigue running at 80% of the ventilatory threshold (treadmill) and in a treadmill running time to exhaustion, respectively.

### 3.5. Effect of Different Characteristics of Studies on Exercise in Which Aerobic Metabolism Is Predominant (≥5 min). Meta-Analysis

Table 6 shows more details about the influence of different characteristics of included studies. Concerning dosage, significant greater results were noticed when supplementation was ≥30 × 10^9^ CFU (*n* = 6) but not when dosage <30 × 10^9^ CFU (*n* = 6) (SMD, 0.47; CI, 0.04 to 0.89; *p* < 0.05; and SMD, 0.20; CI, −0.05 to 0.45; *p* = 0.12, respectively). Regarding supplementation duration, 4 weeks or less (*n* = 6) seems to have significantly greater benefits, but longer periods of supplementation (*n* = 6) did not show any significant improvements (SMD, 0.44; CI, 0.05 to 0.84; *p* < 0.05 and SMD, 0.19; CI, −0.08 to 0.47; *p* = 0.16, respectively). Referring to probiotics type, significant differences were observed when single strain probiotics (*n* = 7) were used in comparison with placebo (SMD, 0.33; CI, 0.06 to 0.60; *p* < 0.05). However, no differences were noticed in the multi strain group (*n* = 5) (SMD, 0.26; CI, −0.08 to 0.60; *p* = 0.14).

Concerning the sex subgroup, significantly better results were observed in the studies conducted in males (*n* = 8); no significant improvements were found in studies performed in females and in studies mixing both sexes (*n* = 3) (SMD, 0.30; CI, 0.04 to 0.56; *p* < 0.05 and SMD, 0.30; CI, −0.19 to 0.79; *p* = 0.23, respectively). Other analyses noted that probiotics improved significantly the results of tests performed to exhaustion (SMD, 0.45; CI, 0.03 to 0.86; *p* < 0.05). Nonetheless, no significant benefits were observed in tests measuring VO_2max_ (SMD, 0.21; CI, −0.11 to 0.52; *p* = 0.21).

### 3.6. Results of Meta-Regression

The meta-regression model revealed that none of the covariates were able to significantly predict the effects of probiotic supplementation on SMD in a highly trained population (duration, *p* = 0.286 and dose, *p* = 0.113). The regression models are shown in Figure 6 and Figure 7.

## 4. Discussion

The main objective of this systematic review and meta-analysis was to analyze and summarize the current literature about the effects of probiotic supplementation on exercise in which aerobic metabolism is predominant (≥5 min) in a highly trained population. The main results showed that probiotics supplementation offers a small and significant positive effect on tests with aerobic metabolism predominance in comparison with placebo. Furthermore, probiotics supplementation effects seem to be better when the dose is ≥30 × 10^9^ CFU, the supplementation period is ≤4 weeks, the probiotic used contains a single strain, the athletes are males, and the outcome of the test is “until reach the fatigue threshold”. These results could be influenced by the type of sport, strain type, dosage, and supplementation duration, and participants’ characteristics (i.e., age, sex, health status, nutrition).

### 4.1. Effect of Probiotics on Exercise in Which Aerobic Meatabolism Is Predominant (≥5 min)

The results of this meta-analysis showed a small but significant positive effect on exercise in which the aerobic metabolism is predominant. In endurance sports, the capacity to maintain the specific intensity for a long period of time becomes essential [50]. The intensity or exercise in which the aerobic metabolism is predominant is equal to or less than VO_2max_ [22]_._ In this context, when exercise intensity is 65% of the VO_2max_, fatty acids provided 50% of the energy substrate, while the remaining 50% was obtained from carbohydrates (CHO) [51]. Thereby, the oxidative metabolism plays a key role in this type of activity [18].

Endurance athletes in particular suffer the so-called leaky gut, which involves symptoms such as vomiting and diarrhea, stomach and intestinal cramps, and nausea due to an increment in GI permeability through the epithelial wall [23,52]. Probiotics could enhance gut-barrier function, by inducing synthesis and assembly of tight junction proteins, and could also prevent disruption of tight junctions produced by injurious factors [53]. Disruptions in tight junctions allow the release of lipopolysaccharides (LPS), which modulate monocyte and macrophage activity and increase the release of pro-inflammatory cytokines [54]. In skeletal muscle cells, circulating LPS could drive the activation of Toll-Like Receptors 4 and 5 (TLR-4 and TLR-5, respectively), and promote Nuclear Factor Kappa light chain enhancer of activated B cell (NF-κB) pathway activation, reduce insulin sensitivity, and enhance protein catabolism and inflammatory cytokine production [55]. There are several factors that could lead to GI disturbance during endurance exercise, such as splanchnic oxidative stress, hypoxia, mechanical stress, hyperthermia induced by exercise, and malabsorption of CHO [56,57]. This reduction in CHO absorption could be considered a limiting factor for performance in endurance exercise lasting more than 60 min [58]. Due to the limited stores of muscle and liver glycogen, oral ingestion of CHO before and during exercise improves performance and could reduce fatigue [59].

Concerning to CHO oxidation, one study measured the capacity of 4 weeks of a multi strain probiotic supplementation (25 billion CFU) to increase the absorption and oxidation of orally ingested maltodextrin solution during 2 h of cycling at 55% VO_2max_ [25]. Results showed a small increase in peak oxidation rates of ingested maltodextrin and mean total CHO oxidation in probiotic group, while a reduction in fat oxidation was observed. Higher plasma glucose and insulin concentrations in the probiotic group suggested a higher duodenal absorption [25]. This is in accordance with a recent study assessing the influence of 4 weeks of probiotic supplementation in systemic metabolism during a marathon [58]. In this research, a greater decrease in some glucogenic amino acids (particularly alanine and arginine) and greater increase in 3-hydroxybutyrate, which is elevated in ketosis conditions, were reported in placebo group, which suggested a shift to lipid metabolism and increased amino acids use as a source of glucose production. The authors hypothesized that the ability of probiotics to maintain intestinal integrity could lead to maintenance of CHO absorption and oxidation during prolonged exercise. Therefore, CHO availability is essential to improve endurance exercise performance, and reduce skeletal muscle turnover and recovery process [58,60].

Regarding probiotic influence in protein metabolism, probiotic intake has been related to improved protein utilization [61], potentially due to an optimization of gut microbiota composition and increasing proteolytic activity [24]. A pilot study carried out in physically active males measured the influence of a multi strain (5 × 10^9^ CFU of *Lactobacillus paracasei* LP-DG and 5 × 10^9^ CFU of *Lactobacillus paracasei* LPC-S01) probiotic in amino-acid absorption of a plant protein (pea protein) [62]. The authors noticed a significant increase in methionine, histidine, valine, leucine, isoleucine, tyrosine, total Branched Chain Amino Acids (BCCA) and total essential amino-acid concentrations in the probiotic group. These findings were corroborated by in vitro analysis [62]. This is in accordance with other studies in which 2 weeks of 20 g of casein were administered with or without 1 billion of *Bacillus coagulans* GBI-30, 6086 [61]. In this study, a probiotic along with 20 g of casein intake significantly increased perceived recovery 24 and 72 h, and muscle soreness 72 h after resistance exercise, both measured by visual analogue scales. *Bacillus coagulans* produce digestive enzymes (proteases), which could facilitate protein digestion and therefore lead to a better absorption and favored muscle-recovery process [61]. In addition, *L. casei* could downregulate some genes involved in the ubiquitin/proteasome pathway, which are implicated in the release of pro-inflammatory signals by NF-κB [63].

Moreover, probiotic intake produced SCFAs through the fermentation of CHO that have not been completely digested in the intestine [26]. It is considered that SCFAs could provide the source up to 10% of total daily energy demands [64]. In addition, SCFAs could exhibit beneficial effects on the host metabolism by modulating epigenetic regulation [65] Gene expression is regulated by the modulation of histone acetylation by histone acetyltransferases and histone deacetylases. SCFAs, by inhibiting histone deacetylase activity, could modulate gene expression [65].

Fatty Acid Receptor 2 (FFA2) and Fatty Acid Receptor 3 (FFA3) are SCFAs receptors [66,67]. FFA2 is expressed in intestinal endocrine L-cells and in adipose tissues [65,66]. SCFAs, by activating FFA2, promote GLP-1 secretion in the gut and suppressed fat accumulation in adipose tissue, leading to an increase in insulin sensitivity [65] and therefore glucose uptake.

Besides, FFA3 is abundantly expressed in sympathetic ganglia and endocrine L-cells. Propionate could activate FFA3 increasing energy expenditure through sympathetic activation. FFA3 triggers PYY gut secretion, which reduces gut motility and thereby could improve nutrient absorption [66].

Furthermore, mitochondrial respiration at the cellular level could be increased by Butyrate (belonging to the group of SCFAs) [27]. Mitochondria are the primary energy centers that process nutrients to produce Adenosine Triphosphate (ATP) [68]. Butyrate could augment the expression of Peroxisome Proliferator-Activated Receptor-Gamma Coactivator 1-alpha (PGC-1α), Peroxisome Proliferator-Activated Receptor delta (PPAR-∂) and Carnitine Palmitoyltransferase-1b (CPT1b), stimulating mitochondrial function [27]. SCFAs (N-butyrate and acetate [69]) could reduce PPAR-γ expression, leading to an increase in mitochondrial UCP2 expression and AMP/ATP ratio [18]. This activates AMPK in liver, adipose tissue [70] and muscle tissue [71], which stimulates glucose uptake, mitochondrial Fatty Acid β-oxidation (FAO) and Oxidative Phosphorylation (OXPHOS) and decreases protein and lipid synthesis [18].

In addition, butyrate could have the ability to increase muscle fiber type I ratio by increasing PGC-1α through an inhibition of the histone deacetylase function [27]. In that way, Chen et al. showed that *L. plantarum* TWK10 increased significantly the number of type I fibers in the gastrocnemius muscle in mice [72].

On the other side, intense training might overproduce ROS due to increased muscle effort [73]. ROS may oxidase proteins, alter their structure, impair their function and affect genetic transcription [74]. Thereby, excessive ROS production could cause a decrement in muscle force generation during repeated contractions and lead to muscle inflammatory diseases [75]. In this context, probiotics could palliate ROS negative effects [18]. Specifically, *Lactobacillus plantarum*, *Lactobacillus gasseri*, *Lactobacillus fermentun*, *Lactobacillus Lactis* and, *Streptoccus thermophilus* could be capable of decreasing ROS through an increase of the antioxidant enzyme, superoxide dismutase (SOD) activity [76]. In addition, *Lactobacillus rhamnosus* IMC 501^®^ and *Lactobacillus. paracasei* IMC 502^®^ could also increase plasma antioxidant levels and neutralized ROS generation after high-intensity exercise [8]. A recent systematic review and meta-analysis observed a slight significant increase in TAC level and a slight significant reduction in MDA levels after probiotic supplementation among adult subjects [20]. A subgroup analysis according to sex showed a significant reduction in MDA levels in both sexes, while MDA level did not significantly decrease in females, which is in accordance with the results observed in this systematic review and meta-analysis. Moreover, the improvement in intestinal homeostasis, including the absorption process, could improve the absorption of antioxidants, increasing the availability of these substances [8]. Some bacteria are able to process polyphenols in the intestine and improve their absorption [77]. Probiotic antioxidant effects are also linked to the synthesis of antioxidant substances such as vitamins B1, B5 and B6 [78]. Thus, probiotics could exert an antioxidant effect and reduce ROS-induced muscle injury [72].

Additionally, probiotics could also affect sports performance, modulating the immune system and enhancing athletes’ health [4]. This improvement is mainly associated with an URTI reduction. Probiotics can reduce URTIs by their capacity to activate T- and B-Lymphocytes, increasing the secretion of Interferon-gamma (IFN-γ), Immuno-globulin A (IgA), and IL-10 cytokines and suppressing the expression of pro-inflammatory cytokines (TNF-α, IL-6, and IL-8) [13]. A recent meta-analysis showed that probiotic supplementation positively affects IL-6 and TNF-α levels. It also revealed a lower total symptom severity score of respiratory infections after probiotics intake, especially when single strain probiotics were consumed [17]. A reduction in URTIs and immune system improvements by probiotic supplementation could increase continuity in training, and therefore influence sports performance [79]. Further, these anti-inflammatory properties could be correlated with a reduction in depression levels, and this reduction could impact athletes’ mental health [80].

All the physiological mechanisms explained above (improve GI barrier function and nutrients metabolism, increase SCFA production, enhance mitochondrial function, palliate ROS overproduction and inflammatory response), influence aerobic metabolism and may explain the results of this meta-analysis (SMD = 0.29).

### 4.2. Effects of Different Characteristics of Studies on Exercise with Aerobic Metabolism Is Predominant (≥5 min)

To get a health benefit, the probiotic definition needs the administration of “adequate amounts” [1]. However, although there is no indication of what that quantity should be, everything seems to indicate that there is a response dose. In this sense, some studies have suggested that there is a response dose in supplementation with probiotics in diarrhea associated with antibiotics [81] and that for blood pressure higher doses of probiotics (> 10^11^ CFU) were more effective than the lowest doses [82]. These results are in accordance with the results of the present meta-analysis where it was shown that probiotics supplementation with doses ≥30 × 10^9^ CFU could lead to increased performance. These effects could be due to a higher probiotics dose and could lead to a greater gut colonization, and therefore, enhance their effects [83,84,85].

Probiotics need time to achieve their key objectives [86]. The time of adaptation of the organism to the probiotics effects is approximately 14 days; this is the period required to adapt GI tract to the administered microorganism [12]. A previous review determined that 10 to 14 days of probiotic supplementation are needed to produce substantial changes in the microbiota [87]. The same review suggested that probiotic supplementation could be more beneficial when using short periods compared to longer periods. Moreover, it defines studies of 4 weeks of duration as short-term studies. Although the exact time is not known, it was suggested that around 4 weeks was needed to induced health benefits for athletes [13,28,29,38,88]. This is also in accordance with the results observed in the present meta-analysis.

Single strain supplementation achieved significant benefits. It is thought that multi train probiotics could improve strains’ GI-tract adhesion [13]. Nevertheless, in multi strain probiotics the dose of each strain could be lower than probiotics using a single strain, which could lead to a reduction in the supplementation effectiveness. This could be a reason for the higher effects obtained with single strain probiotics. A recent systematic review and meta-analysis measuring the effectiveness of probiotic supplementation on respiratory infection and inflammatory biomarkers in elite athletes observed that the total symptom severity score was mainly affected by single strain supplements [17].

Significant differences concerning sexes were found in subgroup analysis. These could be related to the results observed in the study conducted by West et al., in which a difference was found in the number and duration of illness self-reported symptoms between males and females [34]. It is known that there is an immunological difference between sexes [89,90,91,92], which could affect probiotics effects in the immune system [93,94], and thereby performance. However, a pilot study indicated sex as a minor factor in modulating probiotics effects on the immune system [95]. Thus, consensus information about physiological differences between females and males respecting probiotics intake is lacking. To the authors’ knowledge, this is the first meta-analysis carrying out this sex comparison, because it is impossible to compare these results with other studies. Future research is needed to confirm whether probiotics affect males and females differently and to determine the physiological reason for these results. 

In addition, some bacteria present in the gut could play a key role in gut–brain communication due to the generation of some neuroactive molecules and, therefore, improve time to exhaustion during a strenuous test [73]. Several *Lactobacillus* genus strains demonstrated production of γ-aminobutyricacid (GABA), the most important inhibitory transmitter in the brain. Other bacteria have shown the ability to synthetize noradrenaline, dopamine, and serotonin [96]. Dopamine was demonstrated to be synthetized in the GI tract during stressful situations [73]. Moreover, probiotics intake could palliate the reduction in circulating tryptophan (Trp) due to exercise [39]. It is suggested that a greater amount of Trp could lead to an improvement in Trp transport into the brain, supporting serotonin metabolism, which could affect training adherence and performance by influencing individuals’ sensation of fatigue [97]. Thereby, some bacteria have the potential to influence neurotransmitter activity and thus interact with the nervous system to regulate mental health, metabolism, and exercise capacity [73].

Regarding VO_2max_, athletes in the included studies with years of specialized training have relatively high and stable VO_2max_ values [30]. In this context, exercise training seems to be more effective than nutritional strategies for the improvement in the VO_2max_ index [30], mainly due to mitochondrial biogenesis [18]. The training, diet, and recovery of the individuals in some of these studies could be optimal enough to mask any small additional benefits [4].

According to the meta-regression analysis, there were no significant or predictive changes.

### 4.3. Limitations, Strengths and Future Research

Consideration of several limitations should be made when interpreting these results. First and foremost are the different independent variables (probiotic strains, timing, dose, duration, type of sport, and tests) used by authors to examine the effectiveness of probiotics. The fact of using so many diverse outcome tests and supplementation protocols requires conversion to a standardized effect size. With the aim of reducing these limitations, this meta-analysis used a strong statistical analysis and followed a rigorous methodology to analyze and quantify the outcomes. Furthermore, the results should be taken with caution due to the small number of studies included in this systematic review (n=15) and meta-analysis (*n* = 12). This is the first systematic review and meta-analysis assessing the effects of probiotics on exercise in which aerobic metabolism is predominant in a highly trained population. Future studies with similar supplementation protocols and measurement methodologies are needed in order to understand the effect of each strain on sports performance.

### 4.4. Practical Applications

Athletes commonly use ergogenic aids with the aim of maximizing performance. In recent years, probiotics have been investigated for many different purposes that include sports performance. In this systematic review and meta-analysis, the dose used varied from 1.0 × 10^9^ CFU to 4.5 × 10^10^ CFU, and supplementation methods included capsules, powder sachets, and tablet and drinks intake. Furthermore, it is essential to keep in mind that the effects produced are strain- and dose-dependent. This meta-analysis showed that probiotic supplementation increases performance on exercise with aerobic metabolism predominance. In addition, for 4 weeks or less, a dose ≥30 × 10^9^ CFU and single strain probiotics appears to be the optimal form of supplementation. Therefore, this ergogenic aid could be of interest for athletes, coaches, nutritionists, and practitioners in order to optimize sports performance. However, caution should be applied when interpreting these results. Although the number of studies regarding probiotic supplementation and sports performance is increasing, it is still limited in trained populations.

## 5. Conclusions

This systematic review and meta-analysis showed that probiotic supplementation exerts a positive effect on performance with aerobic metabolism predominance in a trained population. The observed results suggest that when the supplementation period was ≤4 weeks and single strain probiotics were consumed, greater benefits could be obtained. Significant improvements were also observed when the supplementation dose was ≥30 × 10^9^ CFU. In addition, males seemed to obtain greater benefits for probiotic supplementation, and probiotic effects appear to be better for tests performed to exhaustion. Nonetheless, results of the meta-regression revealed that any of the measured factors (dose and duration) did not predict probiotics’ effects.

## Figures and Tables

**Figure 1 nutrients-14-00622-f001:**
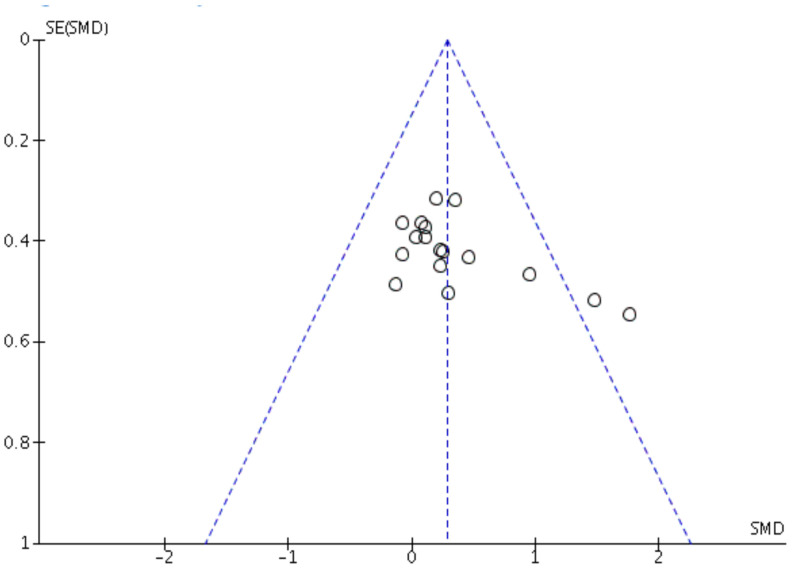
Funnel plot of standard error of sports performance test data by Hedges’ g.

**Figure 2 nutrients-14-00622-f002:**
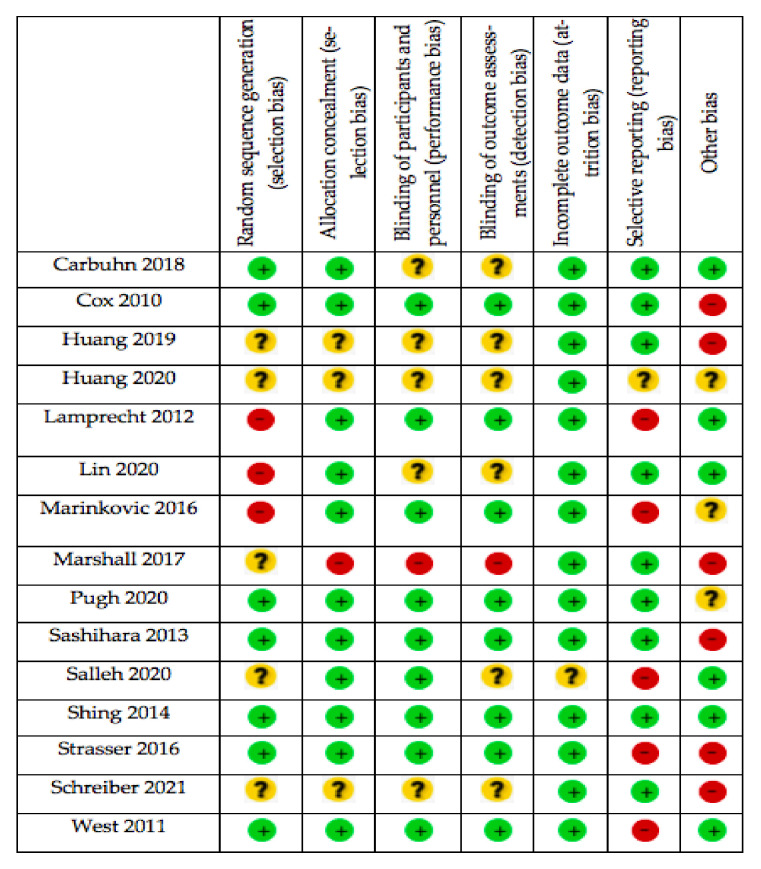
Summary of all risk of bias items. 
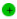
 low risk of bias. 
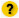
 unknown risk of bias. 
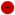
 high risk of bias.

**Figure 3 nutrients-14-00622-f003:**
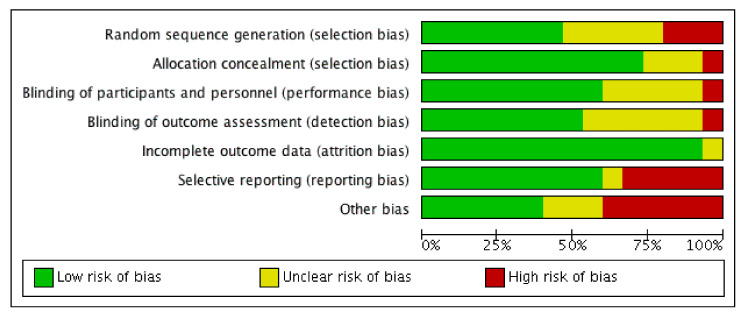
Risk of bias graph expressed as percentages.

**Figure 4 nutrients-14-00622-f004:**
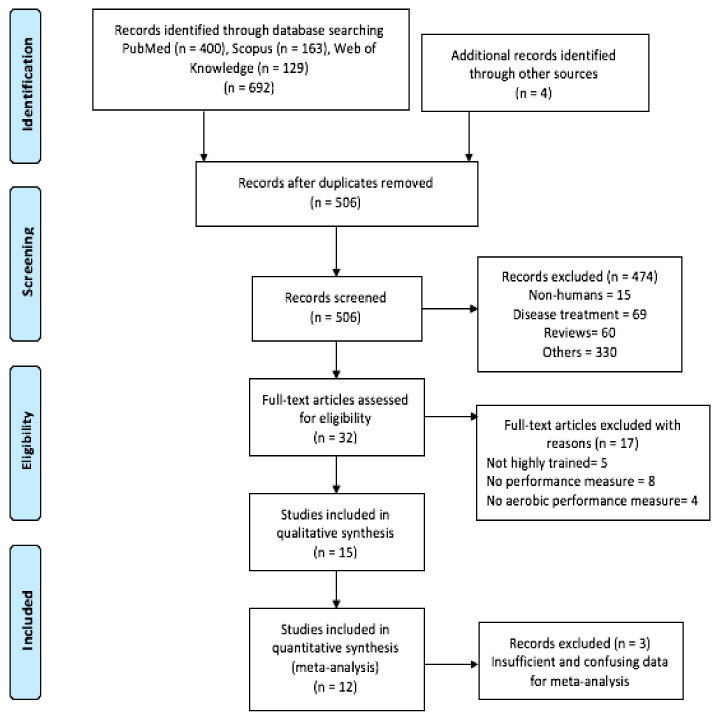
Preferred Reporting Items for Systematic Reviews and Meta-Analyses (PRISMA) flow diagram with information about search and screening process.

**Figure 5 nutrients-14-00622-f005:**
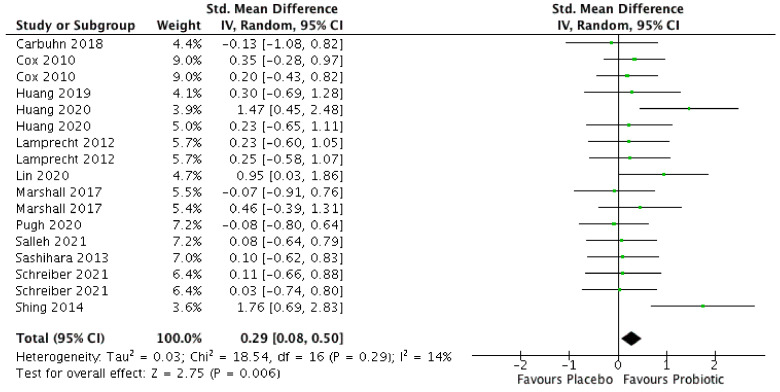
Forest plot performed with Revman comparing the effects of probiotic supplementation on tests in which the aerobic metabolism is predominant.

**Figure 6 nutrients-14-00622-f006:**
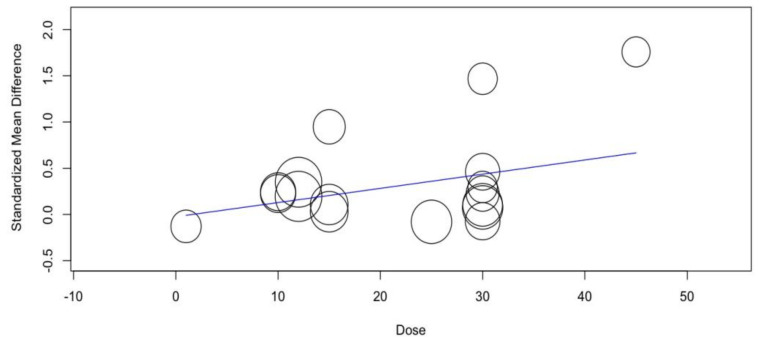
Results of the dose variate random-effect meta-regression for standardized mean differences (SMDs) of tests with aerobic metabolism predominance in a highly trained population.

**Figure 7 nutrients-14-00622-f007:**
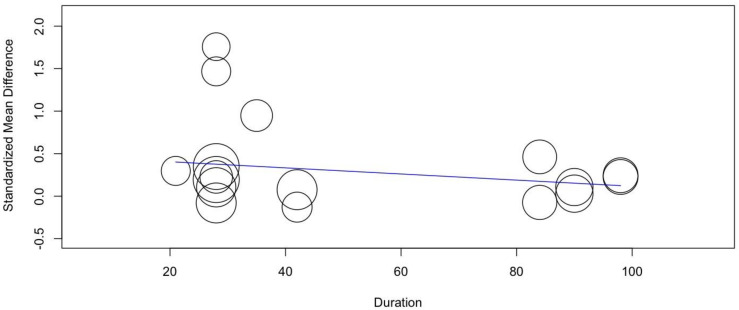
Results of the duration variate random-effect meta-regression for SMDs of tests with aerobic metabolism predominance in a highly trained population.

**Table 1 nutrients-14-00622-t001:** Inclusion and exclusion criteria following each point of PICOS.

	Inclusion Criteria	Exclusion Criteria
P (Population): “athletes and/or Division I and/or trained population (≥8 h/week and/or ≥5 workouts/week)”	Participants had to be athletes and/or Division I and/or trained population (≥8 h/week and/or ≥5 workouts/week clinical trial	Participants had previous health problems or injuries leading to drug intake
I (Intervention) “effects of probiotic supplementation on test with predominance on aerobic metabolism”	Clear information concerning supplementation administration	Unclear information concerning probiotic supplementation
C (Comparators) “similar experimental conditions in the placebo or control group compared with the probiotic group”	-	-
O (Outcome) “performance test with aerobic metabolism dominance”	Used test in which aerobic metabolism is primary	-
S (Study design): “double-blind controlled clinical trial”	Well-designed experiment, a clinical trial, peer-reviewed and original articles written in the English Language; and clear information about funding sources	-

Legend: P, Population; I, Intervention; C, Comparators; O, Outcome; S, Study design.

**Table 2 nutrients-14-00622-t002:** Physiotherapy Evidence Database (PEDro) scale assessment for the included studies according to numbers.

Study	1	2	3	4	5	6	7	8	9	10	11	TOTAL
Carbuhn et al., (2018) [35]	Yes	1	1	0	0	0	0	1	1	1	1	6
Cox et al., (2010) [28]	Yes	1	1	1	1	0	1	1	1	1	1	9
Huang et al., (2019) [29]	Yes	0	0	1	0	0	0	1	1	1	1	5
Huang et al., (2020) [30]	Yes	0	1	0	0	0	0	1	1	1	1	5
Lamprecht et al., (2012) [5]	Yes	0	0	1	0	0	0	1	1	1	1	5
Lin et al., (2020)[36]	Yes	0	0	1	0	0	0	1	1	1	1	5
Marinkovic et al., (2016) [31]	Yes	0	0	1	0	0	0	1	1	1	0	4
Marshall et al., (2017) [37]	Yes	1	0	1	0	0	0	1	1	1	1	6
Pugh et al., (2020) [25]	Yes	0	1	1	1	1	1	1	1	1	1	9
Salleh et al., (2021) [32]	Yes	0	1	1	1	1	1	1	1	1	1	9
Sashihara et al., (2013) [33]	Yes	1	1	1	1	1	1	1	1	1	0	9
Schreiber et al., (2021) [40]	Yes	0	0	0	0	0	0	1	0	1	1	3
Shing et al., (2014) [38]	Yes	1	1	1	1	1	1	1	1	1	1	10
Strasser et al., (2016) [39]	Yes	1	1	1	1	1	1	1	1	1	0	9
West et al., (2011) [34]	Yes	1	1	1	1	1	1	1	1	0	0	8

**Table 3 nutrients-14-00622-t003:** Summary of the studies included in the systematic review that investigated the effect of single strain probiotics on exercise in which aerobic metabolism is predominant (≥5 min).

Author/s	Population	Supplementation protocol	Duration	Training protocol	Test	Outcomes	Effect
Carbuhn et al., (2018) [35]	17 female swimmers from Division I	1 × 10^9^ CFU of *Bifidobacterium longum* 35624 daily (1 capsule per day)	42 days	8–20 h/week, 5 times a week	-500 m freestyle aerobic swim test	-Time trial (s)	- ↔
Cox et al., (2010) [28]	20 highly trained distance male runners (27.3 ± 6.4 years)	1.2 × 10^10^ CFU of *Lactobacillus fermentum* VRI-003 PCC daily (3 capsules twice a day)	28 days	8.2 ± 2.8 h/week endurance training	-Treadmill running test	-Treadmill time (min)	- ↔
						-VO_2max_ (mL/kg/min)	- ↔
Huang et al., (2019) [29]	16 triathletes (Ni) (EG: 22.3 ± 1.2 years; PLA: 20.1 ± 0.3 years)	3 × 10^10^ CFU of *Lactobacillus plantarum* PS128 daily (1 capsule twice a day)	21 days	Specialized training	-VO_2max_ endurance cycling test (48 h after a triathlon championship)	-Time trial (s)	- ↑
Huang et al., (2020) [30]	20 male triathletes (EG: 21.6 ± 1.3 years; PLA: 21.9 ± 1.4 years)	3 × 10^10^ CFU of *Lactobacillus plantarum* PS128 daily (1 capsule twice a day)	28 days	Usual training	-Treadmill running test	-Treadmill time (s)	- ↑
						-VO_2max_ (mL/kg/min)	- ↔
Lin et al., (2020)[36]	21 (14 males and 7 females, aged 20–30 years) well-trained runners	1.5 × 10^10^ CFU of OLP-01, a human strain probiotic derived the *Bifidobacterium longum subsp. Longum* (3 capsules per day)	35 days	Usual training	-12-min running/walking distance	-Distance (m)	- ↑
Marinkovic et al., (2016) [31]	39 male and females’ elite athletes (EG: 23.5 ± 2.7 years; PLA: 22.8 ± 2.5 years)	2 × 10^10^ CFU of *Lactobacillus helveticus* Lafti^®^ L10 daily (capsules)	98 days	>11 h/week	-Graded cardiopulmonary test (treadmill)	-VO_2max_ (mL/kg/min)	- ↔
						-Time (min)	- ↔
Salleh et al., (2021) [32]	30 males badminton players (18–30 years)	3 × 10^10^ CFU of *Lactobacillus casei* daily (commercial probiotic drink) mixed with commercial orange juice (in total 200 mL)	42 days	Usual training	-20 m multi-stage shuttle run	-VO_2max_ (mL/kg/min)	- ↑
Sashihara et al., (2013) [33]	29 male soccer players (EG: 19.8 ± 0.9 years; PLA: 20.2 ± 1.1 years)	3 × 10^10^ CFU of heat-killed cells of *Lactobacillus gasseri* OLL2809 daily (2 tablets 3 times a day)	28 days	Minimum of 5 days/week high intensity training	-Cycle ergometer exercise (1 h at 70% of heart rate reserve)	-Workload (kW/h)	- ↔
West et al., (2011) [34]	88 male and female cyclists (EG: 35.2 ± 10.3 years; PLA: 36.4 ± 8.9 years)	1 × 10^9^ CFU of *Lactobacillus fermentum* VRI-003 PCC^®^ daily (1 capsule per day)	77 days	Usual training	-Incremental performance test (cycle ergometer)	-VO_2max_ (mL/kg/min)	- ↔

Legend: ↔ The effect of probiotic supplementation was not statistically different from placebo; ↑↓ the effect of probiotic supplementation was statistically different (higher and lower, respectively) from placebo; CFU, colony-forming units; EG, experimental group; kilowatts/hour; m, meters; min, minutes; ml/kg/min, milliliters/kilogram/minute; Ni, no information; PLA, placebo group; s, seconds; VTh, Ventilatory threshold; VO_2max_, maximal oxygen consumption; W/kg, Watts/kilogram; kW/h.; W, Wat.

**Table 4 nutrients-14-00622-t004:** Summary of the studies included in the systematic review that investigated the effect of multi strain probiotics on exercise in which the aerobic metabolism is predominant (≥5 min).

Author/s	Population	Supplementation Protocol	Duration	Training Protocol	Test	Outcomes	Effect
Lamprecht et al., (2012) [5]	23 endurance trained men (EG: 37.6 ± 4.7 years; PLA: 38.2 ± 4.4 years)	10^10^ CFU of *Bifidobacterium lactis* W51, *Bifidobacterium bifidum* W23, *Enterococcus faecium* W54, *Lactobacillus brevis* W63, *Lactobacillus acidophilus* W22 and *Lactococcus lactis* W58 daily (2 powder sachets twice a day)	98 days	Usual training	-Triple cycle step test ergometry	-VO_2max_ (mL/kg/min)	- ↔
						-Performance (W/kg)	- ↔
Marshall et al., (2017) [37]	22 male and female marathon runners (EG: 25–50 years; PLA: 23–60 years)	1 × 10^9^ CFU of *Lactobacillus acidophilus* CUL-60 [NCIMB 30157], 9.5 × 10^9^ of *Bifidobacterium bifidum* CUL-20 [NCIMB 30172], 1 × 10^9^ CFU of *Lactobacillus acidophilus* CUL-61 [NCIMB 30156], 0.5 × 10^9^ of *Bifidobacterium animalis subspecies lactis* CUL-34 [NCIMB 30153] and 55.8 mg. d-1 fructooligosaccharides daily (1 capsule per day)	84 days	Not reported	-Graded exercise test to exhaustion (treadmill)	-VO_2max_ (mL/kg/min)	- ↔
					-Marathon des Sables	-Time to completion (min)	- ↔
Pugh et al., (2020) [25]	7 male trained cyclists (23 ± 4 years)	2.5 × 10^10^ CFU of *Lactobacillus acidophilus* (CUL60), *Bifidobacterium bifidum* (CUL20), *Lactobacillus acidophilus* (CUL21), and *Bifidobacterium animalis subsp. Lactis* (CUL34) daily (1 capsule per day)	28 days	Usual training	-120 min of cycling at 55% W_max_	-VO_2max_ (mL/kg/min)	- ↔
Schreiber et al., (2021) [40]	27 male elite cyclists (19–49 years)	15 × 10^9^ CFU of (≥) ≥4.3 × 10^9^ CFU *Bifidobacterium animalis ssp. lactis* Lafti B94 (28.6 %), ≥4.3 × 10^9^ CFU *Lactobacillus helveticus* Lafti L10 (28.6 %), ≥2.1 × 10^9^ CFU *Bifidobacterium longum* R0175 (14.3 %), ≥3.9 × 10^9^ CFU *Enterococcus faecium* R0026 (25.7 %), and ≥0.4 × 10^9^ CFU *Bacillus subtilis* R0179 (2.8 %) (1 capsule per day)	90 days	Usual training	-Time to fatigue (85% maximal power)	-Time to fatigue (min:s)	- ↔
					-Graded exercise test to exhaustion (cycle ergometer)	-VO_2max_ (mL/kg/min)	- ↔
Shing et al., (2014) [38]	10 male runners (27 ± 2 years)	4.5 × 10^10^ CFU of *Lactobacillus rhamnosus, Lactobacillus acidophilus, Lactobacillus casei, Lactobacillus fermentum, Lactobacillus plantarum, Bifidobacterium bifidum*, *Bifidobacterium breve, Bifidobacterium lactis,* and *Streptococcus thermophilus* daily (1 capsule per day)	28 days	Not reported	-Time-to-fatigue run at 80 % of ventilatory threshold (treadmill)	-Time to fatigue (s)	- ↑
Strasser et al., (2016) [39]	29 male and female athletes (EG: 25.7 ± 3.5 years; PLA: 26.6 ± 3.5 years)	1 × 10^10^ CFU of *Bifidobacterium lactis* W51, *Bifidobacterium bifidum* W23, *Lactobacillus acidophilus* W22, *Lactobacillus Brevis* W63 and *Lactococcus lactis* W58 daily *Enterococcus faecium* W54 (1 sachet per day)	84 days	Usual training	-Exercise test untilexhaustion (cycle ergometer)	-VO_2max_ (mL/kg/min)	- ↔

Legend: ↔ The effect of probiotic supplementation was not statistically different from placebo; ↑↓ the effect of probiotic supplementation was statistically different (higher and lower, respectively) from placebo; CFU, colony-forming units; EG, experimental group; kilowatts/hour; m, meters; min, minutes; ml/kg/min, milliliters/kilogram/minute; PLA, placebo group; s, seconds; VTh, Ventilatory threshold; VO_2max_, maximal oxygen consumption; W/kg, Watts/kilogram; kW/h.; W, Wat.

**Table 5 nutrients-14-00622-t005:** Information about study design and nutritional control in included studies.

Study	Study Design	GRADE Approach	Food Record	Prohibited Foods and Supplements
Carbuhn et al., (2018) [35]	RD-BP-C	High	3 days dietary food record	Nutritional supplements Ergogenic supplements Antibiotics and anti-inflammatories
Cox et al., (2010) [28]	RD-BP-CC	High	-	Yoghurt Yoghurt based products
Huang et al., (2019) [29]	D-B	High	-	Fermented food products Probiotics Prebiotics Vitamins Materials and herbal extracts Antibiotics
Huang et al., (2020) [30]	D-BP	High	Dietary record (undefined).	Fermented food Probiotics Prebiotics Alcohol and smoking Antibiotics
Lamprecht et al., (2012) [5]	RD-BP-C	High	7 days food record	-
Lin et al., (2020)[36]	D-B	High	-	-
Marinkovic et al., (2016) [31]	RD-BP-CP	High	-	Yoghurt Fermented milk products Supplements for enhancing the immune system
Marshall et al., (2017) [37]	RIM	Moderate	-	Any other supplements
Pugh et al., (2020) [25]	RD-BP-CC	High	24 h food record	Probiotic foods Alcohol Spicy food Caffeine
Salleh et al., (2021) [32]	RP-C	High	3 days dietary record	Other additional probiotic supplements
Sashihara et al., (2013) [33]	RD-BP-CP	High	-	-
Schreiber et al., (2021) [40]	RD-BP-C	High	Liquid or solid food consumed	Probiotcs supplements Ergogenic supplements Antibiotics and medications
Shing et al., (2014) [38]	RD-BP-CC	High	-	Probiotic supplements Antibiotics and non-steroidal anti-inflammatory drugs
Strasser et al., (2016) [39]	RD-BP-C	High	3 days food record	Fermented dairy products Probiotics Dietary supplements Minerals Vitamins Alcohol Medicines
West et al., (2011) [34]	RD-BP-CP	High	4 days food record	Probiotic enriched yoghurt Supplements/foods containing probiotics Foods or supplements fortify with prebiotics Antibiotics

Legend: D-B, Double-Blind; D-BP, Double-blind, parallel-group; GRADE, Grades of Recommendation, Assessment, Development and Evaluation; RD-BP-C; Randomized Double-Blind Placebo-Control; RD-BP-CC, Randomized Double-Blind Placebo-Control Crossover; RD-BP-CP, Randomized Double-Blind Placebo-Control Parallel; RP-C, Randomized, placebo-controlled; RIM, Randomized independent measures.

**Table 6 nutrients-14-00622-t006:** Different characteristics of studies included in the meta-analysis regarding probiotic effects on exercise with aerobic metabolism predominance.

Subgroups	SMD	95% CI	*p* Value
Dose (CFU)			
<30 × 10^9^ (*n* = 6)	0.20	−0.05 to 0.45	0.12
≥30 × 10^9^ (*n* = 6)	0.47	0.04 to 0.89	<0.05 *
Duration			
≤4 weeks (*n* = 6)	0.44	0.05 to 0.84	<0.05 *
>4 weeks (*n* = 6)	0.19	−0.08 to 0.47	0.16
Strain			
Multi strain (*n* = 5)	0.26	−0.08 to 0.60	0.14
Single strain (*n* = 7)	0.33	0.06 to 0.60	<0.05 *
Sex			
Males (*n* = 8)	0.30	0.04 to 0.56	<0.05 *
Females + mix (males + females) (*n* = 3)	0.30	−0.19 to 0.79	0.23
Test			
To fatigue (*n* = 7)	0.45	0.03 to 0.86	<0.05 *
VO_2max_ (*n* = 7)	0.21	−0.11 to 0.52	0.21

Legend: CFU, colony forming units; CI, confidence interval; n, number of studies; SMD, standardized mean difference (Hedges’ g); VO_2max_, maximal oxygen consumption;* Significantly difference (*p* < 0.05).

## Data Availability

No new data were created or analyzed in this study. Data sharing is not applicable to this article.

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
