# Peer review of "Effects of Probiotic Supplementation on Exercise with Predominance of Aerobic Metabolism in Trained Population: A Systematic Review, Meta-Analysis and Meta-Regression"

_nutrients, 2022, doi:10.3390/nu14030622_

Round 1

Reviewer 1 Report

ABSTRACT

Lines 17-20: Eventually the PICOS must be better defined in the aim of the study. Example: “trained populations” which populations? Any? Trained based on which criteria? Competitive?.

Line 22: In accordance with PRISMA 2020 for abstracts, eligibility criteria, risk of bias, and synthesis of results are missing in the methods reported on the abstract.

Line 28: In accordance with PRISMA 2020 for abstracts, the information about the total number of included studies and participants and summarising relevant characteristics of studies are missing.

INTRODUCTION

Generally well-written and with logic and solid rationale. Minor suggestions:

Line 96: add some examples of how difficult is to standardize or individualize the prescription supplementation.

Line 99: it would be nice to read a statement of contribution to provide some motivation and scientific contribution and practical need for supporting the production of this systematic review. Moreover, it is important to introduce the main outcomes that will be further considered in the results.

METHODS

Lines 114/115: Trained in competitive and/or recreational environments?

Line 116: any dose for the supplementation intake?

Line 122: would be interesting to add as supplementary file the code lines for search in each database

Line 135: it would be nice to see the inclusion and exclusion criteria following each point of PICOS. Possible a table.

The following sections/information is missing:

Data items: List and define all outcomes for which data were sought. Specify whether all results that were compatible with each outcome domain in each study were sought (e.g. for all measures, time points, analyses), and if not, the methods used to decide which results to collect AND List and define all other variables for which data were sought (e.g. participant and intervention characteristics, funding sources). Describe any assumptions made about any missing or unclear information.

RESULTS

Line 223: add the reason for each of the excluded articles (from the 17 excluded in the full-text screening). Maybe add a description point by point in the text or eventually a table as supplementary material.

Line 227: add information, article by article, about which information was missing and caused the exclusion of the 3 studies.

Table 2: critical information is missing. It is important to describe the type of study (parallel and/or crossover) and if the nutrition and other supplementations were controlled for each group (experimental and control).

The funnel plot is missing. It is important to add.

Certainty of evidence is missing. Add the GRADE approach.

DISCUSSION

Generally well-written. Discussion about the certainty of evidence must be done.

Author Response

We really appreciate your concerns, Please see the attachment to observed our comments. 

Reviewer 2 Report

(1) In this paper, the relationship between probiotics and nutrient metabolism is mentioned. Is there any correlation between the content of diet and exercise performance in these collated studies?

(2) In the paper, males seem to obtain greater benefits for probiotic supplementation and effect appear to be better for test performed to exhaustion. Please elaborate on the point of view and provide more information.

 (3) The authors describe that this systematic review and meta-analysis supported the potential effects of probiotics supplementation to improve performance in test in which aerobic metabolism is predominant in trained population. Please provide  more information about its pharmacological evidence and mechanism.

Author Response

(The authors gave the same response as above.)

Reviewer 3 Report

I focused my review on the statistical methods. To that point, the methods utilized are appropriate and I commend the authors for the well performed meta analysis. My concern revolves around what makes this analysis different from the referenced 2021 meta analysis?

Author Response

(The authors gave the same response as above.)

Round 2

Reviewer 1 Report

The article was improved and can be accepted in its current form.